# Creatinine to Body Weight Ratio Is Associated with Incident Diabetes: Population-Based Cohort Study

**DOI:** 10.3390/jcm9010227

**Published:** 2020-01-15

**Authors:** Yoshitaka Hashimoto, Takuro Okamura, Masahide Hamaguchi, Akihiro Obora, Takao Kojima, Michiaki Fukui

**Affiliations:** 1Department of Endocrinology and Metabolism, Graduate School of Medical Science, Kyoto Prefectural University of Medicine, Kyoto 602-8566, Japan; y-hashi@koto.kpu-m.ac.jp (Y.H.); d04sm012@koto.kpu-m.ac.jp (T.O.); michiaki@koto.kpu-m.ac.jp (M.F.); 2Department of Gastroenterology, Asahi University Hospital, Gifu 501-0223, Japan; a-obora@murakami.asahi-u.ac.jp (A.O.); tkojima-gi@umin.ac.jp (T.K.)

**Keywords:** creatinine, diabetes, epidemiology, muscle mass, skeletal muscle mass

## Abstract

We investigated the association between creatinine to body weight (Cre/BW) ratio and incident diabetes. In this cohort study, 9659 men and 7417 women were follow up mean (SD) 5.6 (3.5) years and 5.4 (3.4) years, respectively. For men, tertile 1 (T1; n = 3176), Cre/BW < 0.01275; tertile 2 (T2; n = 3258), 0.01275 ≤ Cre/BW < 0.0148; and tertile 3 (T3; n = 3225), Cre/BW ≥ 0.0148; and for women, T1 (n = 2437), Cre/BMI < 0.0118; T2 (n = 2516), 0.0118 ≤ Cre/BMI < 0.014; and T3 (n = 2477), Cre/BMI ≥ 0.014. Among them, 362 men and 102 women developed diabetes. The hazard ratios (HRs) of incident diabetes in the T2 group was 0.56 (95% CI 0.44–0.71, *p* < 0.001) in men and 0.61 (0.38–0.99, *p* = 0.045) in women and in the T3 group was 0.42 (0.32–0.54, *p* < 0.001) in men and 0.55 (0.34–0.89, *p* = 0.014) in women after adjusting for covariates, compared with the T1 group. Moreover, Δ0.001 incremental of Cre/BW is negatively associated with incident diabetes (adjusted HR 0.84, 95% CI 0.80–0.88, *p* < 0.001 for men and 0.88, 0.81–0.96, *p* = 0.003 for women). In conclusion, Cre/BW ratio is inversely related to incident diabetes. Checking Cre/BW ratios may predict future diabetes risks.

## 1. Introduction

People with type 2 diabetes are increasing and reached 10 million in 2016 in Japan [1]. Hence, it is an important task for us to prevent and treat type 2 diabetes. Previous studies revealed that obesity is a one of the reasons why type 2 diabetes is rapidly increasing [2,3,4].

Diabetes accelerates the reduction of muscle mass by hyperglycemia, insulin resistance, and inflammatory cytokines [5]. In addition, a recent study showed that the degree of glucose tolerance impairment has a close association with decreased lean body mass in non-diabetic women [6]. Therefore, muscle mass is also an important prevention and treatment target for type 2 diabetes.

A pervious study revealed that weight-adjusted appendicular skeletal muscle mass, defined as skeletal muscle mass/body weight, is a risk of type 2 diabetes [7]. In addition, non alcoholic fatty liver disease (NAFLD) [8,9,10,11] and nonalcoholic steatohepatitis [12,13,14], which are closely associated with type 2 diabetes, have an association with weight-adjusted appendicular skeletal muscle mass. In contrast, serum creatinine (Cre), which is known as a marker of kidney function, is influenced by muscle size since muscle mass creates Cre. Serum Cre is reported to be associated with total skeletal muscle mass [15,16,17]. In addition, we recently revealed that creatinine to body weight (BW) (Cre/BW) ratio is associated with a risk of incident NAFLD [18]. Therefore, we hypothesized that Cre/BW also would be associated with incident type 2 diabetes. Here, we investigated the association between Cre/BW ratio and incident type 2 diabetes in this population-based historical cohort study.

## 2. Materials and Methods

### 2.1. Study Patients

The NAGALA (NAfld in Gifu Area, Longitudinal Analysis) study is a historical cohort study of individuals who underwent a medical health check-up program at Asahi University Hospital (Gifu, Japan) starting from 1994 [19]. This medical program is very famous in Japan and is intended to not only find chronic illness and their risks, but also contribute to the health promotion. This center annually received more than 8000 medical examiners. Among the examiners, about 60% of them repeat the program once or twice a year. After obtaining written informed consent, masking personal identifiable information was performed and the medical data of the individuals was stored in a database. For this present analysis, we extracted the data of the individuals who underwent the medical check-up during the years from 2004 to 2014 and who did not have diabetes at the first examination. We set the primary endpoint of this study as the incident of diabetes. The Asahi University Hospital ethics committee permitted this study (approval number 2018-09-01) and this study was performed according to the Declaration of Helsinki. Exclusion criteria were as follows: no follow-up examination, missing data of covariates (including high-density lipoprotein (HDL) cholesterol, BW, and lifestyle factors), and serum creatinine levels over 1.2 mg/dL and over 1.0 mg/dL for men and women, indicating renal dysfunction, were also excluded [20]. In addition, fasting plasma glucose was ≥6.1 mmol/L was also excluded [2] because these participants had a high risk of incident diabetes.

### 2.2. Data Collection and Measurements

Body mass index (BMI) was calculated; BMI (kg/m^2^) = BW (kg)/height2 (m^2^). A standardized self-administered questionnaire was used to obtain the data of lifestyle factors, including alcohol, smoking habits, physical activity, and medical history. Alcohol consumption was measured by estimating mean ethanol intake according to the type and quantity of alcohol consumed (/week) over the past one month. The participants divided into four groups according to the alcohol consumption: none to minimal, <40 g/week; light, 40–140 g/week; moderate, 140–280 g/week; or heavy alcohol consumption, >280 g/week [21]. The participants were also divided into three groups by smoking status: never smoker, ex-smoker, or current smoker. To categorize participants into a non- or regular exerciser, we investigated the participants’ recreational and sports activities. We defined regular exercisers as participants who played any type of sports >1×/week regularly [22]. HbA1c ≥6.5%, fasting plasma glucose ≥7 mmol/L [23], or self-report, including usage of medication for diabetes, were defined as type 2 diabetes. We defined the onset as the time when diabetes was first identified. Standard enzymatic methods were used for serum Cre. We calculated Cre/BW as Cre divided by BW. Insulin resistance was evaluated by triglycerides to high-density lipoprotein (HDL) cholesterol ratio (TG/HDL) [24] and estimated glucose infusion rate (EGIR) [25].

### 2.3. Statistical Analysis

We used EZR (Saitama Medical Center, Jichi Medical University, Saitama, Japan), a graphical user interface for R (The R Foundation for Statistical Computing, Vienna, Austria) [26] and JMP version 13.0 software (SAS Institute Inc., Cary, NC, USA) for the statistical analyses. Mean (standard deviation (SD)) was used for continuous variables and number was used for categorical variables. Because the case of diabetes differed between sex and the distribution of Cre/BW differed between sex, we investigated the following statistical analyses in men and women separately. A *p* value < 0.05 was set for statistical significance. We classified the participants into three groups according to tertile of Cre: For men, tertile 1 (T1), <0.85 mg/dL; tertile 2 (T2), 0.86 ≤ Cre ≤ 1 mg/dL; and tertile 3 (T3), Cre ≥ 1 mg/dL; and for women, T1, Cre ≤ 0.6 mg/dL; tertile 2 (T2), 0.61 ≤ Cre ≤ 0.7 mg/dL; and tertile 3 (T3), Cre ≥ 0.71 mg/dL. Moreover, we classified the participants into three groups according to tertile of Cre/BW ratio: For men, tertile 1 (T1), Cre/BW < 0.01275; tertile 2 (T2), 0.01275 ≤ Cre/BW < 0.0148; and tertile 3 (T3), Cre/BW ≥ 0.0148; and for women, T1, Cre/BW < 0.0118; T2, 0.0118 ≤ Cre/BW < 0.014; and T3, Cre/BW ≥ 0.014. One-way ANOVA and Tukey’s Honest Significant Difference test were used for continuous variables and Pearson’s chi-squared test were used for categorical variables. Kaplan–Meier analysis and log-rank tests were performed to evaluate the difference among the groups according to Cre or Cre/BW ratio. In the log-rank test analyses, a Bonferroni correction was used and set a *p* value <0.0167 to be statistically significant. Further, to examine the effect of the Cre/BW ratio, analysis using a Cox hazard model adjusting for age, fasting plasma glucose, alcohol consumption, exercise, and smoking at baseline examination was conducted for both categorized and continuous variables on the incident diabetes. The association between TG/HDL and EGIR and Cre/BW ratio were evaluated by Pearson’s correlation coefficient. Because there is a close association between Cre/BW and BMI, waist circumference or eGFR (BMI, *r* = −0.58, *p* < 0.001 for men and *r* = −0.54, *p* < 0.001 for women; waist circumference, *r* = −0.58, *p* < 0.001 for men and *r* = −0.56, *p* < 0.001 for women; and eGFR, *r* = −0.58, *p* < 0.001 for men and *r* = −0.61, *p* < 0.001 for women), we did not use these factors as covariates. Proportional hazards assumption was examined by EZR.

## 3. Results

In this study, we included 25,890 participants (14,947 men and 10,943 women; Figure 1). Among them, 7517 participants (4166 men and 3351 women) did not receive follow-up examination, 16 participants (15 men and one woman) had missing data of covariates, and 181 participants (163 men and 18 women) had renal dysfunction. In addition, to keep proportional hazards, we excluded 1100 participants whose fasting plasma glucose was ≥6.1 mmol/L (944 men and 156 women). Thus, this study used 17,076 participants (9659 men and 7417 women).

Table 1 represented the baseline characteristics of the study participants according to the Cre/BW ratio. Age of T1 was older than the other groups in both men and women. BMI of T1 was bigger than the other groups in both men and women, and Cre of T1 was lower than the other groups in both men and women. Moreover, the baseline metabolic parameters (glucose, blood pressure, and cholesterol) of the T1 group were worse than in the other groups in both men and women. In addition, TG/HDL and EGIR were associated with Cre/BW ratio (TG/HDL, *r* = −0.13, *p* < 0.001 for men and *r* = −0.12, *p* < 0.001 for women and EGIR, *r* = 0.33, *p* < 0.001 for men and *r* = 0.32, *p* < 0.001 for women).

During mean (SD) 5.6 (3.5) years follow-up in men and 5.4 (3.4) years follow-up in women, 362 men and 102 women were newly diagnosed with diabetes. The cumulative incidence rate of incident diabetes was 5.2% (incident n/total n = 171/3275) for men and 1.9% (47/2437) for women in T1, 3.3% (incident n/total n = 105/3159) for men and 1.1% (27/2515) for women in T2, and 2.7% (incident n/total n = 86/3325) for men and 1.1% (27/2465) for women in T3.

Figure 2 represents the results of Kaplan–Meier analysis according to the Cre tertiles; there was no statistically significant difference among Cre tertiles in both men and women.

Figure 3 shows the results of Kaplan–Meier analysis according to the Cre/BW ratio tertiles. The proportion of incident diabetes in the T1 group was higher than those in the other groups (all *p* < 0.001).

Table 2 shows the results of Cox hazard model of Cre/BW ratio on incident diabetes. Compared with the T1 group, the hazard ratio (HR) of incident diabetes in the T2 group was 0.56 (95% CI 0.44–0.71, *p* < 0.001) in men and 0.61 (95% CI 0.37–0.98, *p* = 0.042) in women and that in the T3 groups was 0.42 (95% CI 0.32–0.54, *p* < 0.001) in men and 0.53 (95% CI 0.32–0.85, *p* = 0.008) in women in model 1. Furthermore, Cre/BW ratio was negatively associated with incident diabetes (HR of Δ0.001 incremental of Cre/BW, 0.84 (95% CI 0.80–0.88, *p* < 0.001 in men and HR of Δ0.001 incremental of Cre/BW, 0.88 95% CI 0.81–0.96, *p* = 0.003) in women) in model 2.

## 4. Discussion

In this large-scale cohort study, we examined the association between Cre or Cre/BW ratio and incident diabetes and showed that decreased Cre/BW, but not Cre, is associated with incident diabetes in both sexes.

The possible explanations for the relationship between Cre/BW ratio and incident diabetes are as follows. It is well known that during hyperinsulinemia-euglycemia status, muscle mass takes up 80%–90% of glucose in the blood [27]. Lower muscle mass can have reduced capacity of glucose uptake from the blood [6]. Moreover, low muscle mass and inciden diabetes can be mediated by insulin resistance, a key pathogenic mechanism of diabetes [28]. It is well known that height-adjusted SMI, which defined as appendicular skeletal muscle mass/height^2^, is an important marker for sarcopenia [29]. However, heavier weight leads to muscle mass increase, regardless of fat mass [8]. Thus, the proportion of muscle mass per body weight is important. In fact, not height-adjusted SMI, but weight-adjusted appendicular skeletal muscle mass, is associated with cardiometabolic risk factors and insulin resistance [30,31,32,33,34]. Moreover, weight-adjusted appendicular skeletal muscle mass is associated with incident diabetes [7], metabolic syndrome [35], and NAFLD [36,37,38]. This is because the low weight-adjusted appendicular skeletal muscle mass is associated with increased visceral fat. Increasing visceral fat is associated with incident diabetes though the increasing of inflammatory cytokines [39]. In fact, Cre/BW ratio was associated with TG/HDL and EGIR, which are known markers of insulin resistance, in this study. Taking these findings together, Cre/BW ratio is associated with incident diabetes.

Although serum Cre was not associated with incident diabetes in this study population, previous studies have shown that low Cre is associated with the presence of [40,41,42,43] or incident diabetes [44,45,46,47]. These studies showed that the risk of incident diabetes in the lowest serum Cre group was higher than that in highest serum Cre group [44,45,46,47]. However, a previous study also showed that there is a nonlinear relationship between serum Cre and incident diabetes [45]. On the other hand, there is a linear relationship between serum Cre/BW and incident diabetes in this study. This might be because that although there is a close association between Cre and Cre/BW, the data is dissociated in some cases. These dissociated cases were an important target in incident diabetes. Thus, the reason why the result about serum Cre level and incident diabetes in this study was different from the previous studies might be that the ratio of dissociated cases was different.

Several strengths of this study exist, including the relatively large number of participants included. However, this study has limitations as well. First, there is a possibility of underestimating cases of diabetes because we did not have the data of an oral glucose tolerance test. However, the rate of incident diabetes was almost same as the previous meta-analysis of the Japanese population [48]. In addition, a familiarity for type 2 diabetes mellitus might be a source of bias. Second, this study included Japanese individuals only, thus, whether our study results generalize to non-Japanese populations, especially non-Asian populations, is uncertain and further studies might be needed. Third, we did not have data on insulin levels, thus, the association between insulin resistance, such as homeostasis model assessment insulin resistance and Cre/BW ratio, was unclear. However, we showed the association between Cre/BW ratio and TG/HDL and EGIR, which are known as markers of insulin resistance [24,25]. Furthermore, we did not have data of skeletal muscle mass and body fat mass. Fourth, the design of this study was retrospective. Thus, further studies are needed to confirm our results. Moreover, because of nature of the health check program, nearly 40% of participants did not receive a follow-up examination. Thus, this might be a selection bias. However, the clinical backgrounds between the participants with/without follow-up examination was not different (Appendix A). Lastly, the data of some other elements such as diet and insulin levels, which may confound the association between Cre/BW and diabetes, were lacking.

## 5. Conclusions

In conclusion, this study is the first to demonstrate that Cre/BW ratio is negatively associated with incident diabetes in both men and women. These results proposed that a low Cre/BW ratio predicts an increased risk of diabetes.

## Figures and Tables

**Figure 1 jcm-09-00227-f001:**
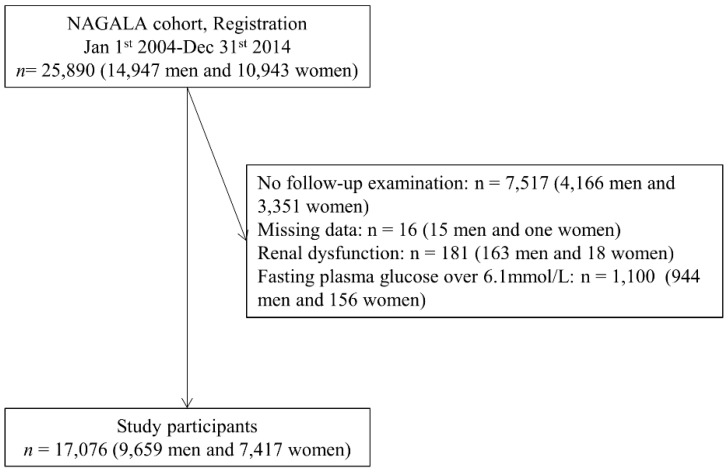
Inclusion and exclusion flow chart.

**Figure 2 jcm-09-00227-f002:**
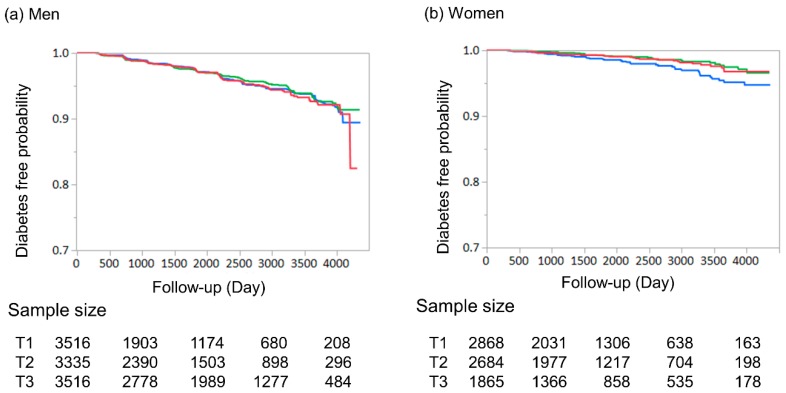
Kaplan–Meier analysis of incident diabetes according to the Cre tertiles. The vertical axis is the diabetes free rate and the horizontal axis is time as months. (**a**) Kaplan–Meier analysis for incident diabetes in men. Red line indicates tertile 1 (Cre < 0.85 mg/dL). Green line indicates tertile 2 (Cre 0.86 ≤ Cre ≤ 1 mg/dL). Blue line indicates tertile 3 (Cre ≥ 1 mg/dL). Log rank test was used to investigate the association among the tertiles of Cre and *p* value was 0.991. (**b**) Kaplan–Meier analysis for incident diabetes in women. Red line indicates tertile 1 (Cre < 0.6 mg/dL). Green line indicates tertile 2 (0.61 ≤ Cre ≤ 0.7 mg/dL). Blue line indicates tertile 3 (Cre ≥ 0.71 mg/dL). Log rank test used to investigate the association among the groups of Cre tertiles. To correct familiar error, Bonferroni correction was used and a *p* value < 0.0167 was considered statistically significant. Tertile 1 vs. Tertile 2, *p* value = 0.45; Tertile 1 vs. Tertile 3, *p* value = 0.087; and Tertile 2 vs. Tertile 3, *p* value = 0.014.

**Figure 3 jcm-09-00227-f003:**
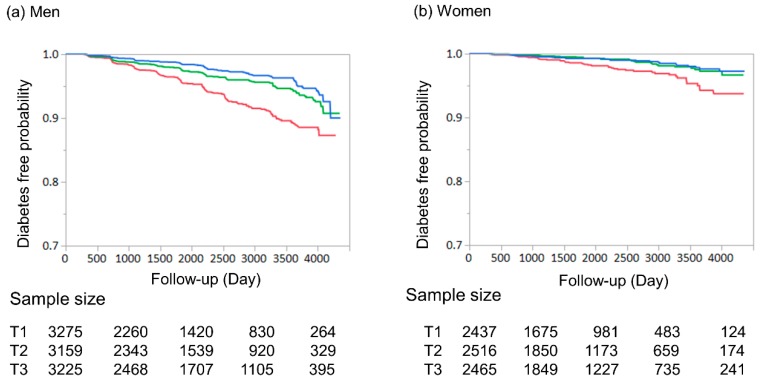
Kaplan–Meier analysis of incident diabetes according to the Cre/BW ratio tertiles. The vertical axis is diabetes free rate and the horizontal axis is time as months. (**a**) Kaplan–Meier analysis for incident diabetes in men. Red line indicates tertile 1 (Cre/BW ratio < 0.01275). Green line indicates tertile 2 (0.01275 ≤ Cre/BW ratio < 0.0148). Blue line indicates tertile 3 (Cre/BW ratio ≥ 0.0148). Log rank test was performed to investigate the association among the groups of Cre/BW ratio. Bonferroni correction was performed to correct familiar error and a *p* value < 0.0167 was considered statistically significant and all *p* values < 0.001. (**b**) Kaplan–Meier analysis for incident diabetes in women. Red line indicates tertile 1 (Cre/BW ratio < 0.0118). Green line indicates tertile 2 (0.0118 ≤ Cre/BW ratio < 0.014). Blue line indicates tertile 3 (Cre/BW ratio ≥ 0.014). Log rank test was performed to investigate the association among the groups of Cre/BW ratio. To correct familiar error, Bonferroni correction was used and a *p* value < 0.0167 was considered statistically significant. Tertile 1 vs. Tertile 2, *p* value = 0.003 and Tertile 1 vs. Tertile 3, *p* value = 0.006 and Tertile 2 vs. Tertile 3, *p* value = 0.854.

**Table 1 jcm-09-00227-t001:** Baseline characteristics of study participants according to the tertiles of creatinine/body weight (Cre/BW) ratio.

**Men**	**ALL,** **n = 9659**	**Tertile 1 (Cre/BW Ratio < 0.01275),** **n = 3176**	**Tertile 2 (0.01275 ≤ Cre/BW Ratio < 0.0148),** **n = 3258**	**Tertile 3 (Cre/BW Ratio ≥ 0.0148),** **n = 3225**	***p***
Age (year)	45.3 (9.3)	43.5 (8.5)	45.4 (9.1) *	47.1 (9.8) *,^†^	<0.001
Body weight (kg)	67.6 (10.0)	75.2 (9.9)	67.0 (7.0) *	60.6 (6.7) *,^†^	<0.001
Height (cm)	170.6 (6.0)	172.9 (5.7)	170.6 (5.6) *	168.1 (5.6) *,^†^	<0.001
Body mass index (kg/m^2^)	23.2 (3.0)	25.1 (3.1)	23.0 (2.3) *	21.5 (2.3) *,^†^	<0.001
Waist circumference (cm)	81.1 (8.0)	86.0 (7.8)	80.7 (6.3) *	76.3 (6.5) *,^†^	<0.001
Fasting plasma glucose (mmol/L)	5.3 (0.4)	5.4 (0.4)	5.3 (0.4) *	5.3 (0.4) *,^†^	<0.001
Hemoglobin A1c (%)	5.2 (0.3)	5.2 (0.3)	5.2 (0.3) *	5.1 (0.3) *,^†^	<0.001
Hemoglobin A1c (mmol/L)	32.9 (3.6)	33.4 (3.6)	32.8 (3.5) *	32.5 (3.6) *,^†^	<0.001
Creatinine (mg/dL)	0.92 (0.12)	0.84 (0.10)	0.92 (0.10) *	1.01 (0.11) *,^†^	<0.001
Creatinine (µmol/L)	81.6 (10.5)	74.6 (8.5)	81.1 (8.4) *	88.9 (9.3) *,^†^	<0.001
Triglycerides (mmol/L)	1.2 (0.8)	1.3 (0.8)	1.2 (0.8) *	1.1 (0.7) *,^†^	<0.001
HDL cholesterol (mmol/L)	1.3 (0.3)	1.2 (0.3)	1.3 (0.3) *	1.4 (0.4) *,^†^	<0.001
Systolic blood pressure (mmHg)	120.2 (14.7)	123.4 (14.3)	120.0 (14.4) *	117.3 (14.8) *,^†^	<0.001
Diastolic blood pressure (mmHg)	76.0 (10.2)	77.8 (10.1)	75.9 (10.0) *	74.3 (10.1) *,^†^	<0.001
Exercise (-/+)	7822/1837	2646/530	2650/608	2526/699	<0.001
Smoking (Non/Past/Current)	3039/3074/3546	995/946/1235	984/1064/1210	1060/1064/1101	<0.001
Alcohol intake (Non-min/light/moderate/heavy)	5727/1505/1317/1110	1937/465/398/376	1900/515/470/373	1890/525/449/361	0.111
Cre/BW ratio	0.014 (0.002)	0.011 (0.001)	0.014 (0.001)	0.016 (0.002)	<0.001
Incident diabetes	362	169	107	86	<0.001
**Women**	**ALL,** **n = 7417**	**Tertile 1 (Cre/BW Ratio < 0.0118),** **n = 2437**	**Tertile 2 (0.0118≤ Cre/BW Ratio < 0.014),** **n = 2515**	**Tertile 3 (Cre/BW Ratio ≥ 0.014),** **n = 2465**	***p***
Age (year)	44.2 (9.2)	43.8 (8.7)	44.1 (9.2)	44.6 (9.5) *,^†^	0.004
Body weight (kg)	52.9 (8.0)	58.6 (8.8)	52.1 (5.9) *	48.0 (5.2) *,^†^	<0.001
Height (cm)	158.1 (5.4)	159.5 (5.3)	158.1 (5.3) *	156.7 (5.3) *,^†^	<0.001
Body mass index (kg/m^2^)	21.2 (3.0)	23.1 (3.5)	20.9 (2.3) *	19.6 (2.0) *,^†^	<0.001
Waist circumference (cm)	72.1 (8.4)	77.5 (8.8)	71.4 (6.5) *	67.4 (6.3) *,^†^	<0.001
Fasting plasma glucose (mmol/L)	5.0 (0.4)	5.1 (0.4)	5.0 (0.4) *	4.9 (0.4) *,^†^	<0.001
Hemoglobin A1c (%)	5.2 (0.3)	5.2 (0.3)	5.2 (0.3) *	5.1 (0.3) *	<0.001
Hemoglobin A1c (mmol/L)	33.2 (3.6)	33.7 (3.7)	33.0 (3.6) *	32.8 (3.7) *	<0.001
Creatinine (mg/dL)	0.68 (0.10)	0.60 (0.08)	0.67 (0.08) *	0.76 (0.09) *,^†^	<0.001
Creatinine (µmol/L)	59.8 (9.0)	53.3 (6.7)	59.2 (6.8) *	66.8 (7.8)*, ^†^	<0.001
Triglycerides (mmol/L)	0.7 (0.4)	0.7 (0.5)	0.7 (0.4) *	0.7 (0.4) *	<0.001
HDL cholesterol (mmol/L)	1.6 (0.4)	1.6 (0.4)	1.6 (0.4) *	1.7 (0.4) *,^†^	<0.001
Systolic blood pressure (mmHg)	110.6 (15.2)	114.4 (15.6)	109.9 (14.7) *	107.7 (14.5) *,^†^	<0.001
Diastolic blood pressure (mmHg)	68.5 (10.2)	70.5 (10.6)	68.0 (10.0) *	67.1 (9.7) *,^†^	<0.001
Exercise (−/+)	6188/1229	2099/338	2108/407	1981/484	<0.001
Smoking (Non/Past/Current)	6396/503/518	2072/192/173	2178/166/171	2146/145/175	0.084
Alcohol intake (Non-min/light/moderate/heavy)	6705/416/211/85	2235/120/56/26	2252/156/78/29	2218/140/77/30	0.209
Cre/BW ratio	0.013 (0.003)	0.010 (0.001)	0.013 (0.001) *	0.016 (0.002) *,^†^	<0.001
Incident diabetes	101	47	27	27	0.013

Cre/BW, Creatinine to body weight; HDL, high density lipoprotein. Continuous variables are expressed as mean (SD), and the differences among tertiles are evaluated by one-way ANOVA and Tukey’s post-hoc tests. Categorized variables are expressed as number, and the differences among tertiles are evaluated by Chi-squared test. * *p* < 0.05, vs. Tertile 1; ^†^
*p* < 0.05, vs. Tertile 2.

**Table 2 jcm-09-00227-t002:** Adjusted hazard ratio of incident diabetes.

**Men**	**Model 1**		**Model 2**	
	**HR with 95% CI**	***p* Value**	**HR with 95% CI**	***p* Value**
Age, years	1.05 (1.03–1.06)	<0.001	1.05 (1.04–1.06)	<0.001
Fasting plasma glucose, 0.1 mmol/L	1.32 (1.28–1.37)	<0.001	1.32 (1.27–1.36)	<0.001
Light drinker	0.65 (0.47–0.90)	0.001	0.66 (0.48–0.91)	0.009
Moderate drinker	0.54 (0.38–0.77)	<0.001	0.55 (0.38–0.77)	<0.001
Heavy drinker	0.92 (0.68–1.24)	0.586	0.95 (0.70–1.26)	0.716
Regular exerciser	0.73 (0.53–0.99)	0.046	0.73 (0.52–0.99)	0.039
Ex-smoker	0.85 (0.63–1.13)	0.263	0.84 (0.62–1.12)	0.231
Current smoker	1.58 (1.22–2.05)	<0.001	1.55 (1.19–2.02)	0.010
Cre/BW ratio tertiles				
T1	Reference	-	-	-
T2	0.56 (0.44–0.71)	<0.001	-	-
T3	0.42 (0.32–0.54)	<0.001	-	-
Cre/BW ratio, 0.001 incremental	-	-	0.84 (0.80–0.88)	<0.001
**Women**	**Model 1**		**Model 2**	
	**HR with 95% CI**	***p* Value**	**HR with 95% CI**	***p* Value**
Age, years	1.05 (1.02–1.07)	<0.001	1.05 (1.02–1.07)	<0.001
Fasting plasma glucose, 0.1 mmol/L	1.40 (1.33–1.48)	<0.001	1.40 (1.32–1.48)	<0.001
Light drinker	0.35 (0.09–0.94)	0.035	0.34 (0.08–0.91)	0.030
Moderate drinker	0.81 (0.20–2.24)	0.720	0.78 (0.19–2.14)	0.667
Heavy drinker	0.64 (0.10–2.12)	0.512	0.69 (0.11–2.26)	0.587
Regular exerciser	0.84 (0.46–1.42)	0.527	0.86 (0.50–1.50)	0.605
Ex-smoker	1.50 (0.62–3.08)	0.341	1.53 (0.63–3.12)	0.319
Current smoker	3.49 (1.85–6.09)	<0.001	3.57 (1.91–6.23)	<0.001
Cre/BW ratio tertiles				
T1	Reference	-	-	-
T2	0.61 (0.37–0.98)	0.042	-	-
T3	0.53 (0.32–0.85)	0.008	-	-
Cre/BW ratio, Δ0.001 incremental	-	-	0.88 (0.81–0.96)	0.003

Cre/BW, Creatinine to body weight. Hazard ratios of light, moderate, and heavy drinkers evaluated with none-to-minimal drinker as reference. Hazard ratio of regular exerciser evaluated non-exerciser as reference. Hazard ratios of ex- and current smokers evaluated with non-smoker as reference. Hazard ratios of T2 (0.0127 ≤ Cre/BW ratio < 0.0148 in men and 0.0118 ≤ Cre/BW ratio < 0.014 in women) and T3 (Cre/BW ratio ≥ 0.0148 in men and Cre/BW ratio ≥ 0.014 in women) of Cre/BW ratio tertiles evaluated with T1 (Cre/BW ratio < 0.01275 in men and Cre/BW ratio < 0.0118 in women) as reference.

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
