# Peer review of "Creatinine to Body Weight Ratio Is Associated with Incident Diabetes: Population-Based Cohort Study"

_jcm, 2020, doi:10.3390/jcm9010227_

Round 1

Reviewer 1 Report

In the study entitled “Creatinine to body weight ratio is associated with incident diabetes: population-based cohort study” the Authors assessed whether the creatinine to body weight (Cre/BW) ratio has a predictive role in the incidence of type II diabetes mellitus.

Overall, the paper is of interest. The sample size fits with the study purpose. I would suggest the following improvements: 

Introduction section (lines 40-42): please clearly state the study purpose . According to the Authors opinion, might the familiarity for type II diabetes mellitus represent a source of bias of the study results? Did the Authors adjust by this factor?

Discussion: overall, this section is poor and needs to be improved

Discussion (line 175): “6”: please revise

Discussion (lines 177-180): the results should not be showed into the Discussion section

Author Response

To Reviewer 1

I give deep thanks for your important and kind review.

Introduction section (lines 40-42): please clearly state the study purpose. According to the Authors opinion, might the familiarity for type II diabetes mellitus represent a source of bias of the study results? Did the Authors adjust by this factor?

Response

Thank you for your comment. A pervious study revealed that weight-adjusted appendicular skeletal muscle mass, defined as skeletal muscle mass/ body weight, is a risk of incident type 2 diabetes. In addition, NAFLD and nonalcoholic steatohepatitis, which are closely associated with type 2 diabetes, have association with weight-adjusted appendicular skeletal muscle mass. In contrast, serum creatinine (Cre) is reported to be associated with skeletal muscle mass. In addition, we recently revealed that creatinine to body weight (BW) (Cre/BW) ratio is associated with a risk of incident NAFLD. Therefore, we hypothesized that Cre/BW also would be associated with incident type 2 diabetes. According to your comment, we have revised the Introduction section and added a reference described as below.

“A pervious study revealed that weight-adjusted appendicular skeletal muscle mass, defined as skeletal muscle mass/ body weight, is a risk of incident type 2 diabetes [7]. In addition, NAFLD [8-11] and nonalcoholic steatohepatitis [12-14], which are closely associated with type 2 diabetes, have association with weight-adjusted appendicular skeletal muscle mass. In contrast, serum creatinine (Cre), which is known as a marker of kidney function, is influenced by muscle size, since muscle mass creates Cre. Serum Cre is reported to be associated with total skeletal muscle mass [15-17]. In addition, we recently revealed that creatinine to body weight (BW) (Cre/BW) ratio is associated with a risk of incident NAFLD [18]. Therefore, we hypothesized that Cre/BW also would be associated with incident type 2 diabetes. Here, we investigated the association between Cre/BW ratio and incident type 2 diabetes in this population-based historical cohort study.”

18. Okamura, T., Hashimoto, Y., Hamaguchi, M., Obora, A., Kojima, T., Fukui, M. Creatinine-to-bodyweight ratio is a predictor of incident non-alcoholic fatty liver disease: A population-based longitudinal study. Hepatol Res 2019 in press. doi: 10.1111/hepr.13429.

In addition, as you say, familiarity for type II diabetes mellitus might be a source of bias and it was desirable to adjust this point. Unfortunately, however, we could not adjust this point.

Thus, we have mentioned this point as an one of the limitations of study in the Discussion section described as below.

“In addition, a familiarity for type 2 diabetes mellitus might be a source of bias.”

Discussion: overall, this section is poor and needs to be improved.

Discussion (line 175): “6”: please revise.

Response

Thank you for your kind suggestion. According to your suggestion. We have revised the Discussion section described as below.

“The possible explanations of the relationship between Cre/BW ratio and incident diabetes are as follows. It is well known that during hyperinsulinemia-euglycemia status, muscle mass takes up 80–90% of glucose in the blood [27]. Lower muscle mass can have reduced capacity of glucose uptake from the blood [6]. Moreover, low muscle mass and incident diabetes can be mediated by insulin resistance, a key pathogenic mechanism of diabetes [28]. It is well known that height-adjusted SMI is an important marker for sarcopenia [29]. However, heavier weight leads to muscle mass increase, regardless of fat mass [8]. Thus, the proportion of muscle mass per body weight is important. In fact, not height-adjusted SMI, but weight-adjusted appendicular skeletal muscle mass, is associated with cardiometabolic risk factors and insulin resistance [30-34]. Moreover, weight-adjusted appendicular skeletal muscle mass is associated with incident diabetes [7], metabolic syndrome [35] and NAFLD [36-38]. This is because that low weight-adjusted appendicular skeletal muscle mass is associated with increased visceral fat. Increasing visceral fat is associated with incident diabetes though increasing of inflammatory cytokines [39]. In fact, Cre/BW ratio were associated with TG/HDL and EGIR, which are known markers of insulin resistance, in this study. Taking these finding together, Cre/BW ratio is associated with incident diabetes.”

Discussion (lines 177-180): the results should not be showed into the Discussion section

Response

Thank you for your suggestion. According your suggestion, the results have descried in the Results section.

Reviewer 2 Report

The paper by Hashimoto et al. represents an interesting large cohort study,  which evaluates an association between the creatinine to body weight ratio and the risk of diabetes development. The study is well designed and represents quite novel findings, of important clinical significance. However, unfortunately due to several language mistakes, the manuscript is very difficult to follow. Authors should consult the professional language editing service to correct their manuscript. The manuscript requires also additional assesment by the journal's statistician for correctness of aplied methods. The discussion section is relatively short and most of this section contains the list of study limitations and strenghts, however lacks extensive comparison of their results with those obtained by other authors. Authors should once again do a literature search for similar studies.

Author Response

To Reviewer 2

I give deep thanks for your important and kind review.

However, unfortunately due to several language mistakes, the manuscript is very difficult to follow. Authors should consult the professional language editing service to correct their manuscript.

Response

Thank you for your suggestion. According to your suggestion, this manuscript has been edited by native English speaker.

The manuscript requires also additional assesment by the journal's statistician for correctness of aplied methods.

Response

Thank you for your suggestion. Corresponding author, Masahide Hamaguchi, is a specialist in statistics.

The discussion section is relatively short and most of this section contains the list of study limitations and strenghts, however lacks extensive comparison of their results with those obtained by other authors. Authors should once again do a literature search for similar studies.

Response

Thank you for your kind suggestion. According to your suggestion. We have revised the Discussion section and added References described as below.

“The possible explanations of the relationship between Cre/BW ratio and incident diabetes are as follows. It is well known that during hyperinsulinemia-euglycemia status, muscle mass takes up 80–90% of glucose in the blood [27]. Lower muscle mass can have reduced capacity of glucose uptake from the blood [6]. Moreover, low muscle mass and incident diabetes can be mediated by insulin resistance, a key pathogenic mechanism of diabetes [28]. It is well known that height-adjusted SMI is an important marker for sarcopenia [29]. However, heavier weight leads to muscle mass increase, regardless of fat mass [8]. Thus, the proportion of muscle mass per body weight is important. In fact, not height-adjusted SMI, but weight-adjusted appendicular skeletal muscle mass, is associated with cardiometabolic risk factors and insulin resistance [30-34]. Moreover, weight-adjusted appendicular skeletal muscle mass is associated with incident diabetes [7], metabolic syndrome [35] and NAFLD [36-38]. This is because that low weight-adjusted appendicular skeletal muscle mass is associated with increased visceral fat. Increasing visceral fat is associated with incident diabetes though increasing of inflammatory cytokines [39]. In fact, Cre/BW ratio were associated with TG/HDL and EGIR, which are known markers of insulin resistance, in this study. Taking these finding together, Cre/BW ratio is associated with incident diabetes.”

“29. Chen, L.K., Liu, L.K., Woo, J., Assantachai, P., Auyeung, T.W., Bahyah, K.S., Chou, M.Y., Chen, L.Y., Hsu, P.S., Krairit, O., Lee, J.S., Lee, W.J., Lee, Y., Liang, C.K., Limpawattana, P., Lin, C.S., Peng, L.N., Satake, S., Suzuki, T., Won, C.W., Wu, C.H., Wu, S.N., Zhang, T., Zeng, P., Akishita, M., Arai, H. Sarcopenia in Asia: consensus report of the Asian Working Group for Sarcopenia. J Am Med Dir Assoc. 2014;15:95–101.

30. Kim, T.N., Park, M.S., Lee, E.J., Chung, H.S., Yoo, H.J., Kang, H.J., Song, W., Baik, S.H., Choi, K.M. Comparisons of three different methods for defining sarcopenia: An aspect of cardiometabolic risk. Sci Rep 2017;7:6491. 

31.Furushima, T., Miyachi, M., Iemitsu, M., Murakami, H., Kawano, H., Gando, Y., Kawakami, R., Sanada, K. Comparison between clinical significance of height-adjusted and weight-adjusted appendicular skeletal muscle mass. J Physiol Anthropol 2017;36:15.

32. Takamura, T., Kita, Y., Nakagen, M., Sakurai, M., Isobe, Y., Takeshita, Y., Kawai, K., Urabe, T., Kaneko, S. Weight-adjusted lean body mass and calf circumference are protective against obesity-associated insulin resistance and metabolic abnormalities. Heliyon 2017;3:e00347.

33. Lim, S., Kim, J.H., Yoon, J.W., Kang, S.M., Choi, S.H., Park, Y.J., Kim, K.W., Lim, J.Y., Park, K.S., Jang, H.C. Sarcopenic obesity: prevalence and association with metabolic syndrome in the Korean Longitudinal Study on Health and Aging (KLoSHA). Diabetes Care 2010;33:1652–1654.

34. Scott, D., Park, M.S., Kim, T.N., Ryu, J.Y., Hong, H.C., Yoo, H.J., Baik, S.H., Jones, G., Choi, K.M. Associations of Low Muscle Mass and the Metabolic Syndrome in Caucasian and Asian Middle-aged and Older Adults. J Nutr Health Aging 2016;20:248–255.

35. Park, B.S., Yoon, J.S. Relative skeletal muscle mass is associated with development of metabolic syndrome. Diabetes Metab J 2013;37:458–464. 

36. Osaka, T., Hashimoto, Y., Okamura, T., Fukuda, T., Yamazaki, M., Hamaguchi, M., Fukui, M. Reduction of Fat to Muscle Mass Ratio Is Associated with Improvement of Liver Stiffness in Diabetic Patients with Non-Alcoholic Fatty Liver Disease. J Clin Med 2019;8:E2175. 

37. Shida, T., Oshida, N., Oh, S., Okada, K., Shoda, J. Progressive reduction in skeletal muscle mass to visceral fat area ratio is associated with a worsening of the hepatic conditions of non-alcoholic fatty liver disease. Diabetes Metab Syndr Obes 2019;12:495–503.

38. Mizuno, N., Seko, Y., Kataoka, S., Okuda, K., Furuta, M., Takemura, M., Taketani, H., Hara, T., Umemura, A., Nishikawa, T., Yamaguchi, K., Moriguchi, M., Itoh, Y. Increase in the skeletal muscle mass to body fat mass ratio predicts the decline in transaminase in patients with nonalcoholic fatty liver disease. J Gastroenterol 2019;54:160–170. 

39. Fontana, L., Eagon, J.C., Trujillo, M.E., Scherer, P.E., Klein, S. Visceral fat adipokine secretion is associated with systemic inflammation in obese humans. Diabetes 2007;56:1010–1013.”

Round 2

Reviewer 2 Report

I wish to thank authors for addressing all my queries.